# Properties of TiAlN Coatings Obtained by Dual-HiPIMS with Short Pulses

**DOI:** 10.3390/ma16041348

**Published:** 2023-02-05

**Authors:** Alexander Grenadyorov, Vladimir Oskirko, Alexander Zakharov, Konstantin Oskomov, Sergey Rabotkin, Vyacheslav Semenov, Andrey Solovyev, Alexander Shmakov

**Affiliations:** 1The Institute of High Current Electronics SB RAS, 2/3, Akademichesky Ave., 634055 Tomsk, Russia; 2Budker Institute of Nuclear Physics SB RAS, 11, Acad. Lavrentieva Pr., 630090 Novosibirsk, Russia

**Keywords:** TiAlN coatings, dual magnetron sputtering, short pulse HiPIMS, wear resistance, heat resistance, synchrotron radiation

## Abstract

The paper focuses on the dual high-power impulse magnetron sputtering of TiAlN coatings using short pulses of high power delivered to the target. The surface morphology, elemental composition, phase composition, hardness, wear resistance, and adhesive strength of TiAlN coatings with different Al contents were investigated on WC–Co substrates. The heat resistance of the TiAlN coating was determined with synchrotron X-ray diffraction. The hardness of the TiAlN coating with a low Al content ranged from 17 to 30 GPa, and its wear rate varied between 1.8∙10^−6^ and 4.9∙10^−6^ mm^3^·N^−1^·m^−1^ depending on the substrate bias voltage. The HF1–HF2 adhesion strength of the TiAlN coatings was evaluated with the Daimler–Benz Rockwell C test. The hardness and wear rate of the Ti_0.61_Al_0.39_N coating were 26.5 GPa and 5.2∙10^−6^ mm^3^·N^−1^·m^−1^, respectively. The annealing process at 700 °C considerably worsened the mechanical properties of the Ti_0.94_Al_0.06_N coating, in contrast to the Ti_0.61_Al_0.39_N coating, which manifested a high oxidation resistance at annealing temperatures of 940–950 °C.

## 1. Introduction

Due to their high hardness, resistance to oxidation, and wear resistance, nitride coatings (such as TiN and TiAlN) are often used in industries, for example, to improve the lifetime and resistance to high-temperature oxidation of cutting tools [1,2,3].

Cathodic arc evaporation (CAE) is one of the most widely used methods to create TiN and TiAlN coatings in industrial conditions [4,5]. The main advantage of CAE is its high deposition rate with a high content of ionizing material in the plasma [6]. A major shortcoming of CAE is its droplet fraction, which has negative effects on coatings’ topography, corrosion and oxidation stability (droplets act as diffusion paths), and tribological properties [4,7,8,9]. Additionally, droplets could contribute to coating degradation through nucleation sites for shear cracks and the intrusion of abrasive fragments into sliding contact [10]. Different techniques are used to reduce the number and size of entrained droplets, such as filtered cathodic vacuum arcs [11,12], steered arcs [13,14], and pulsed arc depositions [15]. Nevertheless, it is difficult to completely avoid the droplet fraction.

Lately, high-power impulse magnetron sputtering (HiPIMS) has been used as an alternative to the abovementioned techniques in depositing nitride coatings based on titanium and aluminum [4]. It was shown that HiPIMS could be used to create coatings with a lower surface roughness than that of coatings generated with arc deposition techniques [6,16,17]. The higher ionization of HiPIMS-deposited materials allows for the formation of a denser structure of titanium nitride [18]. Alhafian et al. [4] compared the properties of TiAlN coatings obtained with HiPIMS and CAE and found that they both provided the coatings with a denser structure and high hardness. However, HiPIMS produced lower surface roughness and friction coefficient values. This method could be used to control the elastic modulus of TiAlN coatings by changing the deposition parameters (pressure and bias voltage), which further affected parameters such as the plasticity index and resistance to plastic deformation correlating with tribological properties [19].

The properties of HiPIMS-deposited coatings strongly depend on pulsed power parameters such as frequency, pulse time and waveform, and duty cycle. Ghailane et al. [20] used HiPIMS with a frequency 500 Hz to synthesize TiN coatings. It was shown that HiPIMS-deposited TiN coatings were harder (29–34 GPa) than those deposited by DC magnetron sputtering (25–27 GPa). This result was explained by the higher (111) texture coefficient, which ranged between 0.85 and 0.95 for HiPIMS and between 0.66 and 0.82 for DC magnetron sputtering. In [21], TiN coatings were deposited by HiPIMS with a single pulse at a pulse time of 4–16 µs and a frequency of 0.8–4 kHz, with three very short pulses of a 4 µs duration with an inter-pulse time of 50 µs and a frequency of 1.5 kHz. In the single-pulse mode, the ionized flux fraction increased from 20 to 45% as the pulse time decreased from 16 to 4 μs. In the multi-pulse mode, the highest ionized flux fraction (up to 52%) was detected. Nevertheless, the pulse time increase was accompanied by increases in the TiN coating’s hardness from 15 to 20 GPa and its elastic modulus. Chang et al. [22] reported that duty cycles varied from 3 to 30% when using a 200 Hz pulse frequency for the HiPIMS of TiN coatings. The hardness of such coatings was 29–33 GPa. It was noted that the growth in the duty cycle from 3 to 30% resulted in higher friction coefficients of 0.31 to 0.63 and wear rates of 3.3 to 6.1·10^−6^ mm^3^·N^−1^·m^−1^ due to the lower hardness, adhesive strength, and increased surface roughness. Alhafian et al. [4] used a two-component target Ti_0.5_Al_0.5_ to create a TiAlN coating at a 500 Hz pulse frequency with three duty cycles of 2.5, 3.5, and 5.5%. The elastic modulus of the coating decreased with increasing numbers of duty cycles from 2.5 to 5.5%, which was accompanied by a hardness change within 27–33 GPa. In [23,24], a HiPIMS-deposited TiAlN coating at a 25 μs pulse time and a 500 Hz frequency possessed a high hardness of 30–35 GPa and withstood repeated heating up to 950 °C without damage or delamination. It was also shown that preliminary plasma treatment and the deposition of the intermediate metallic layer improved the heat resistance of this coating due to its high adhesion level and low defect content.

When manufacturing TiAlN coatings, it is better to employ dual magnetron sputtering systems with a variety of important features. Firstly, they eliminate the problem of anode disappearance during the deposition of non-conductive coatings, thereby providing a long-term and stable deposition process. Secondly, the closed-field configuration of the dual system allows one to increase the current density on the substrate [25], which has a positive effect on coating adhesion, density, and other properties [26,27]. Thirdly, one-component Ti or Al targets and magnetron power control make it possible to vary coating composition over a wide range.

In our previous work [28] devoted to dual-HiPIMS, we developed a power supply with an independent control for bipolar pulse parameters. It was shown that the ion flux on the substrate was higher when depositing metal coatings at a constant average HiPIMS discharge power with short pulses (10–20 μs) [29].

The aim of this work was to synthesize TiAlN coatings using dual-HiPIMS and to study the obtained coatings’ structure, mechanical, tribological, and heat resistance properties.

## 2. Materials and Methods

### 2.1. Preparation of TiAlN Coatings

WC–Co alloy (8 wt.% Co., CNIC, Ningbo, China) samples were used as 13 × 13 × 5 mm substrates. Their surfaces were roughly ground on #600 and #1200 diamond plates, polished with 40/28 and 10/7 µm diamond grinding paste, and ultrasonically cleaned in acetone and isopropyl alcohol for 5 min in each liquid.

The TiAlN coatings were deposited using dual magnetrons. Al and Ti (99.95%) cathodes with a diameter 76 mm were used for sputtering. The proposed dual-HiPIMS vacuum system is schematically illustrated in Figure 1. The vacuum chamber was evacuated to a residual pressure of 8·10^−3^ Pa with a turbo-molecular pump. The substrate was heated up to 500 °C with a ceramic infrared heater. An APEL-M-10HPP-1500 (Applied Electronics, Russia) high-current power supply was used for the magnetrons.

Two series of depositions, differing in Al magnetron power, were carried out to produce coatings with different Al contents. In the first series of experiments, the deposition of TiAlN coatings with the minimum Al content was conducted in a mixture of argon and nitrogen at gas rates of 5 and 0.3 L/h, respectively. The operating discharge parameters of the Ti magnetron included a frequency of 5 kHz, pulse time of 20 µs, voltage of 696 V, and power of 1 kW. Under these conditions, the magnetron with the Al cathode operated at the lowest power of 50 W. Prior to the coating deposition, bipolar voltage with negative pulses of 900 V and 100 kHz and a 70% duty cycle was supplied to the substrate to improve the adhesive strength and enable 5 min of surface activation. Afterwards, the negative pulse amplitude was reduced to 100 V, and the coating deposition was initiated and lasted for 120 min. The positive pulse of the substrate bias was 4 μs, and the peak amplitude was 20% of the negative pulse. Some of TiAlN coatings with the low Al contents were deposited at a floating potential and a substrate bias voltage of −50 V.

In the second series of experiments, the discharge power of the magnetron with the Al cathode was increased for the deposition of the TiAlN coatings with the higher Al content. In the Al magnetron, the discharge power was 1 kW and the voltage was 632 V at the same frequency and pulse duration as in the case of the Ti cathode. The nitrogen gas rate was increased up to 0.7 L/h. The substrate surface was activated as described above. The substrate temperature and biasing (−100 V) were the same as in the case of the first series of experiments.

The oscilloscope patterns in Figure 2 illustrate the typical pulse waveforms of the discharge current and discharge voltage in the dual magnetron system during the co-sputtering of the Ti and Al targets. One can see rectangular discharge current pulses and triangular current pulses.

### 2.2. Investigation Methods

The current density on a substrate steel holder with a diameter of 130 mm was measured with a current shunt with 10 Ohm resistance and a limiter diode. The coating thickness was determined with the Calotest technique using a CAT-S-0000 Calotest machine (CSEM, Neuchâtel, Switzerland).

A NanoTest 600 hardness tester (Micro Materials Ltd., Wrexham, Wales, GB) was used to measure the coating nanohardness with the Oliver and Pharr method [31] at am indentation load of 20 mN, a loading–unloading time of 20 s, and exposure to the maximum load for 10 s. Nanohardness was measured at 20 points, and the obtained results were averaged.

In conformity with the VDI 3198 standard [32], the Rockwell C scale hardness test was conducted with a TK-2 spheroconical diamond indenter (IvMashprom, Yekaterinburg, Russia) for the adhesive strength evaluation of the TiAlN coatings. The adhesive strength was measured at an indentation load of 60 kg (588 N). Indentation causes local elastoplastic deformation around an indent, which can lead to a coating fracture. We propose a correlation between the size and physical form of the elastoplastic deformation area and coating adhesion. This correlation can be used to determine the adhesive strength of a coating. The coating adhesion to the substrate was estimated on a scale from strong (HF1) to poor (HF6) adhesive strength, depending on the delamination level and crack number.

Scanning electron microscopy (SEM) was carried out with a Quanta 200 3D (FEI Company, Hillsboro, OR, USA) microscope equipped with an energy-dispersive X-ray (EDX) analyzer. The accelerating voltage was 30 kV.

XRD patterns of the TiAlN coatings were obtained using a Shimadzu XRD-6000 Diffractometer (Kyoto, Japan). Measurements were conducted using Cu *K*α radiation in a scan range of 30–80° at a scanning step of 0.02°. The analysis of the phase composition and lattice parameters was performed using the PDF4+ database and the PowderCell 2.4 Rietveld program.

The wear rate of the coatings was evaluated using the ball-on-disc method. Testing conditions included a sliding speed of 25 mm/s, sliding distance of 150 m, normal load of 5 N applied to the Al_2_O_3_ ball (diameter of 6 mm) used as a counter body, and wear track diameter of 5 mm. In order to focus on the coating failure, a relatively small normal load of 5 N was used to minimize the coating adhesion effect, provide a sufficiently long coating lifetime, and eliminate friction measurement errors due to the surface roughness of substrates. The wear rate was calculated as the ratio between the wear volume and the product of the normal force and total sliding distance, viz., mm^3^·N^−1^·m^−1^. After testing, the wear track profile was measured with an MNP-2 interference microscope profilometer (Russia) and a 130 contact profilometer (Russia).

The Hertzian maximum pressure (*P_max_*) was calculated using Equations (1)–(3) [33], where *a* is the radius of the contact area, *F* is the normal load applied to the counter body, *E** is the reduced elastic modulus of the friction pair, *R* is the radius of the counter body, *E*_1_ and *E*_2_ are moduli of elasticity, and υ_1_ and υ_2_ are Poisson’s ratios of the counter body and the coating, respectively.
(1)$$a={\left(\frac{3\cdot F\cdot R}{4\cdot E^{*}}\right)}^{1/3}$$
(2)$$E^{*}={\left(\frac{1-\upsilon_{1}^{2}}{E_{1}}+\frac{1-\upsilon_{2}^{2}}{E_{2}}\right)}^{-1}$$
(3)$$P_{max}=\frac{F}{\pi \cdot a^{2}}$$

In these calculations, we assumed the Poisson’s ratio of the TiAlN coating to be 0.177 [34]. The elastic modulus and Poisson’s ratio of the counter body were 382 GPa and 0.24, respectively [35].

Some coatings were annealed in a laboratory electric furnace (OOO “Sikron”, St Petersburg, Russia) in order to compare their heat resistance. The temperature was gradually increased to 700 °C at 150 degrees per hour. When the temperature reached 700 °C, the substrates were cooled at the same rate.

The heat resistance of the TiAlN coating was also investigated on a Zr substrate using synchrotron X-ray diffraction (SXRD) on the VEPP-3 storage ring at the Siberian Synchrotron and Terahertz Radiation Centre of the Budker Institute of Nuclear Physics SB RAS, Novosibirsk, Russia. During this investigation, the substrate was heated from 30 to 1300 °C at an angular velocity of 15 dpm. The synchrotron radiation wavelength was 0.172 nm. The diffraction angle, 2θ, of X-ray radiation was recalculated to the angle at the 0.1541 nm wavelength (Cu *K*α radiation) to compare the obtained results with those measured with a commercial X-ray diffraction apparatus.

## 3. Results and Discussion

### 3.1. TiAlN Coatings Deposited by the Al Magnetron at 30 W of Power

The SEM images in Figure 3 present the surfaces and cross-sections of the TiAlN coatings deposited in the first series of experiments by dual-HiPIMS at a floating potential and −50 and a substrate bias voltage of −100 V. The coating thickness was ~2 µm. One can see that the coating density grew with the increasing bias voltage and, consequently, the energy of bombarding ions. The main effect of increased ion energy is usually a higher mobility of adatoms, accelerated nucleation at the initial stage of the coating growth, and the repeated sputtering of the material deposited at later stages. This is accompanied by changes in the composition, crystal structure, and texture, as well as the evolution of multiple phases [36].

Table 1 summarizes the elementary composition of the TiAlN coatings deposited at different substrate bias voltages. The presence of W and Co elements in the EDX spectra of the TiAlN coatings can be explained by their presence in the substrate. The coating composition differed from the stoichiometric state ((Ti+Al):N = 1:1), and these coatings were under-stochiometric in nature. When the bias voltage grew from floating to −100 V, the (Ti+Al)/N ratio was reduced from 1.58 to 1.24, i.e., it tended to be stoichiometric. The color of the coating changed from dark- to light-yellow. Due to the low Al content, the coating color was similar to that of the TiN coatings.

Unlike the growing N content, the coating’s Ti content decreased with the increasing substrate bias voltage. The Al content insignificantly changed and ranged within 6–8 at.%. The growth in the N content was conditioned by the more effective capture of N atoms in the near-surface regions of the coating and their more intense excitation and ionization, which promoted the chemical reaction of the nitride formation.

Figure 4 plots the deposition rate of the TiAlN coating and substrate current density depending on the bias voltage. The substrate current density grew from 0.83 to 1.17 mA/cm^2^ with the increasing bias voltage. When the bias voltage increased from −20 to −40 V, the current density rapidly grew due to the electron repellence from the substrate, as the bias voltage was greater than the floating potential. When the bias voltage reached −80 V and higher, the substrate current density started to saturate because only ions were collected on the substrate. When the bias voltage increased, the deposition rate lowered from 23 to 16 nm/min, probably due to the coating etching with accelerated ions [37]. On the other hand, it may have been caused by a compaction of the coating structure during the bombardment by heavy metal ions, whose concentration in HiPIMS is high [38].

Table 2 presents the values of hardness *H*, elastic modulus *E*, plasticity index *H*/*E*, and resistance to plastic deformation *H*^3^/*E*^2^ obtained after the analysis of the loading-u–loading curves of the nanoindentation of the TiAlN coatings deposited at different substrate bias voltages (see Figure 5). The increase in the latter up to −100 V resulted in the improvement of the coating’s mechanical properties, i.e., growth was observed in its hardness from 17 to 30 GPa, plasticity index from 0.057 to 0.082, and resistance to plastic deformation from 56 to 202 MPa. A higher bias voltage (>100 V) was not considered here since it led to reductions in the deposition rate, residual stress growth, and formation of radiation defects in the coating structure. The defect concentration in the TiN coatings was observed to be higher when the ion energy increased by more than 100 eV [39].

In the study of Musil and Vlček [40], the *H* and *H*^3^/*E*^2^ values of TiAlN coatings magnetron-sputtered under different deposition conditions ranged within 35–45 GPa and 400–600 MPa, respectively. Thus, the *H* and *H*^3^/*E*^2^ values of the TiAlN coatings obtained in this study with the Al magnetron at 30 W of power were more consistent with the TiN coating parameters. The latter were characterized by a lower hardness and resistance to plastic deformation. The *H*/*E* ratio of the obtained coatings was noticeably lower than the value of 0.1, which is considered the minimum required for hard nitride coatings resistant to cracking [41].

According to Figure 6 and Table 2, the higher bias voltage led to increases in the wear track area, maximum wear depth, and wear rate of the TiAlN coating, though it appears that the higher the hardness, the higher the wear resistance. The increase in the wear rate can probably be explained by the solid particles of the counter body and the coating itself involved in the abrasive friction.

As can be seen in Table 2, the Hertzian maximum pressure of the TiAlN coatings varied from 1.49 to 1.60 GPa, depending on the substrate bias voltage. Morozov and Zernin [42] observed the maximum shear stress of 39 or 40 μm at a depth of ∼0.5*a*. This depth value was many times higher than the coating thickness, which indicated that the nucleation of plastic deformation occurred inside the substrate.

The SEM images in Figure 7 show the indentation craters formed after the Rockwell hardness test on the TiAlN-coated substrate surface. There was no coating delamination, but there were cracks near the indentation crater. Longer cracks could be observed on the substrates covered by the harder TiAlN coating, i.e., obtained at bias voltages of −50 and −100 V. These TiN coatings showed a satisfactory adhesion strength of HF1–HF2 according to the VDI 3198 standard [32].

The XRD patterns shown in Figure 8 were obtained for the TiAlN coatings low in Al and deposited at the substrate floating potential and bias voltages of −50 and −100 V. The coating peaks were related to titanium nitride. One can see that with increasing bias voltage, the (200) TiN peak intensity decreased while the (111) peak intensity grew. The (111) diagonal plane texture prevailed with increasing substrate bias voltage due to the increased energy of adatoms. This plane texture allowed adatoms to occupy more energetically favorable positions in the lattice. Despite the lower surface energy of the (001) plane [43], the film texture evolved toward the (111) plane due to the kinetic limitations associated with the higher cation surface diffusivity on the (001) TiN surface than on the (111) TiN surface [44].

The hardness of the TiN coating with the (111) texture was higher than that of the coating with the other texture [20]. The term texture coefficient is commonly used in the literature, and it is calculated as
(4)$$T_{\left(111\right)}=\frac{I_{\left(111\right)}}{I_{\left(111\right)}+I_{\left(200\right)}},$$
where *I*_(111)_ and *I*_(200)_ are the integral intensities of the (111) and (200) diffraction peaks, respectively.

In our case, the texture coefficient was 0.37 at the substrate floating potential. At substrate bias voltages of −50 and −100 V, the texture coefficient increased to 0.78 and 0.86, respectively.

The lattice parameter *a* of the TiAlN coating deposited at the substrate floating potential was 4.2499 Ǻ, which agreed with the parameter value suggested by the Joint Committee on Powder Diffraction Standards (JCPDS) database, namely, *a* = 4.24173 Ǻ (file N 38-1420). When the substrate bias voltage reached −50 and −100 V, the lattice parameter became 4.2500 and 4.2697 Ǻ, respectively, indicating increased compressive residual stresses in the coatings.

### 3.2. TiAlN Coatings Deposited by the Al Magnetron at 1 kW of Power

Although the TiN coating showed high hardness and low wear coefficient values, its oxidation resistance was low at high (>500 °C) temperatures. This coating cannot therefore be used in tool protection during powerful metalworking at surface temperatures of up to 1000 °C [45,46]. TiAlN coatings with an Al content of 25 to 70 at.% subjected to high-temperature oxidation must be used for powerful metalworking since they generate an Al_2_O_3_ film on their surface, have low friction coefficients, and have a high wear resistance, thus protecting deep coating layers from oxidation [47].

Table 3 shows the elemental composition of the TiAlN coating deposited by dual-HiPIMS using one-component Ti and Al cathodes and the Al magnetron at 1 kW of power and −100 V of substrate bias voltage. The coating’s chemical composition was Ti_0.61_Al_0.39_N. Although the power of the Ti and Al magnetrons was the same (1 kW), the Al content was much lower than the Ti content in the coating. This could be explained by the fact that the formation of non-conducting compounds (nitrides or oxides) was more intensive on the Al target surface than on the Ti target during the reactive sputtering. Non-conductive films on the target surface reduced its sputtering rate.

Depending on the substrate position relative to magnetrons and their discharge power, the Ti or Al content in the coating can vary. In our experiments, the substrate was fixed at similar distances from both cathodes. The Ti_0.61_Al_0.39_N coating hardness was slightly lower than the hardness of the Ti_0.94_Al_0.06_N coating synthesized at a substrate bias voltage of −100 V; the plasticity index and resistance to plastic deformation values were comparable (see Table 4). The greatest advantage of the Ti_0.61_Al_0.39_N coating was its higher thermal stability, which is described below.

SEM images of the Ti_0.61_Al_0.39_N coating surface, cross-section, and hardness indentation are presented in Figure 9. Similar the Ti_0.94_Al_0.06_N coating, the surface of the Ti_0.61_Al_0.39_N coating was smooth, without the droplet fraction. The adhesive strength of this coating was high. The adhesion test conducted in accordance with the VDI 3198 standard [32] showed no delamination, only microcracks, near the indentation crater, thus demonstrating an adhesion strength of HF1–HF2.

### 3.3. Heat-Resistance of TiAlN Coatings

As reported in [1,46,48], TiAlN coatings demonstrate a higher thermal stability than TiN coatings. This was confirmed in this study following the 700 °C annealing of Ti_0.94_Al_0.06_N and Ti_0.61_Al_0.39_N coatings in air followed by their nanoindentation. The Ti_0.61_Al_0.39_N coating showed a higher thermal stability and an insignificant decrease (<10%) in hardness after annealing, as illustrated by the loading–unloading curves shown in Figure 10, which were obtained after the nanoindentation of the Ti_0.94_Al_0.06_N and Ti_0.61_Al_0.39_N coatings before and after annealing at 700 °C.

The Ti_0.94_Al_0.06_N coating demonstrated lower values of hardness (by 54.3%), plasticity index (by 46.3%), and resistance to plastic deformation(by 86.6%). Its elastic recovery *W*_e_ was reduced by 36.7% after annealing. The annealing process provided a significant penetration depth for the indenter, thus indicating the degrading mechanical properties of the Ti_0.94_Al_0.06_N coating. Regarding the Ti_0.61_Al_0.39_N coating, its *H*, *H*/*E*, *H*^3^/*E*^2^ and *W*_e_ parameters were reduced by 9.7, 21.3, 43.3, and 5.5%, respectively.

The 900 °C annealing of the Ti_0.94_Al_0.06_N and Ti_0.61_Al_0.39_N coatings deposited onto WC–Co substrates resulted in the fracture of the latter. Hörling et al. [49] observed surface damage of a WC–Co substrate after annealing at 900 °C due to the stronger diffusion of Co atoms from the deep layers to the surface. Therefore, we used a Zr substrate to study the thermal resistance of the Ti_0.61_Al_0.39_N coating during its dynamic heating up to 1300 °C in air.

Figure 11 shows the phase composition of this coating during heating to 1300 °C in air. In the initial state, a wide, high-intensity reflection was observed for the coating that mostly contained titanium nitride, with aluminum being in the amorphous state. During heating up to 350 °C, this wide reflection shifted toward the low-angle region due to thermal expansion. Within the 350–830 °C temperature range, the lattice parameter of the coating was reduced, which was manifested by the reflection shift to the right. This could have been associated with the formation of the TiAlN crystal phase. At 830 °C, the coating reflection again demonstrated the thermal expansion, whereas at 940–950 °C, new phases appeared; the wide intense reflection remained unchanged except for the thermal expansion. At an annealing temperature of ~1260 °C, the rutile phase (TiO_2_) appeared. The initial reflections of the substrate disappeared and new reflections appeared, thus confirming the formation of the rutile phase.

The SXRD patterns presented in Figure 12 for the Ti_0.61_Al_0.39_N coating were obtained before (1) and after (2) heating to 1300 °C. Before heating, the major phase was the TiN phase, as indicated by the (111) peak, and the Al phase was not detected since it was in the amorphous state. During annealing, the appearance of the TiAlN crystal phase was indicated by the (111) peak shifting toward the large-angle region, unlike the (111) peak of the TiN phase, i.e., Al atoms incorporated into the TiN phase lattice. A (111) TiO_2_ phase peak was observed along with the major TiAlN phase after annealing.

Therefore, the heat resistance of the Ti_0.61_Al_0.39_N coating synthesized with short-pulse dual-HiPIMS was considerably higher than that of the first Ti_0.5_Al_0.5_N coatings obtained in 1986 [50] by the sputter ion plating process, which oxidized in the temperature range of 700 to 750 °C.

## 4. Conclusions

In this work, TiAlN coatings were synthesized for the first time with the short-pulse dual-HiPIMS of one-component Ti or Al targets. The results of this study clearly showed that the substrate bias voltage affected the structure, mechanical, and tribological properties of the dual-HiPIMS-deposited TiAlN coating low in Al. It was found that the growth in the substrate bias voltage up to −100 V improved the coating hardness from 17 to 30 GPa, which matched the growth in the (111) texture coefficient from 0.37 to 0.86. The results indicated that the 700 °C annealing of the Ti_0.94_Al_0.06_N coating provided significant reductions in its mechanical properties, as the hardness was reduced from 30 to 13.7 GPa.

It was possible to synthesize a hard (>25 GPa) Ti_0.61_Al_0.39_N coating using dual-HiPIMS with short pulses and one-component Al and Ti cathodes. After annealing at 700 °C, the mechanical properties of the Ti_0.61_Al_0.39_N coating were not as strongly degraded (from 26.5 to 23.9 GPa) as those of the Ti_0.94_Al_0.06_N coating. The synchrotron X-ray diffraction phase composition analysis of the Ti_0.61_Al_0.39_N coating during dynamic heating up to 1300 °C in air showed that its structural stability remained unchanged until temperatures of ~940–950 °C were reached.

## Figures and Tables

**Figure 1 materials-16-01348-f001:**
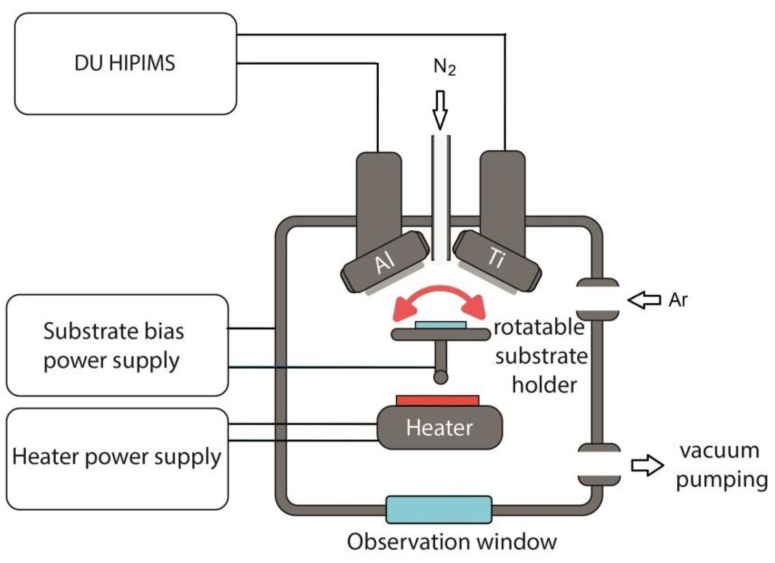
Schematic of the dual-HiPIMS vacuum installation [30].

**Figure 2 materials-16-01348-f002:**
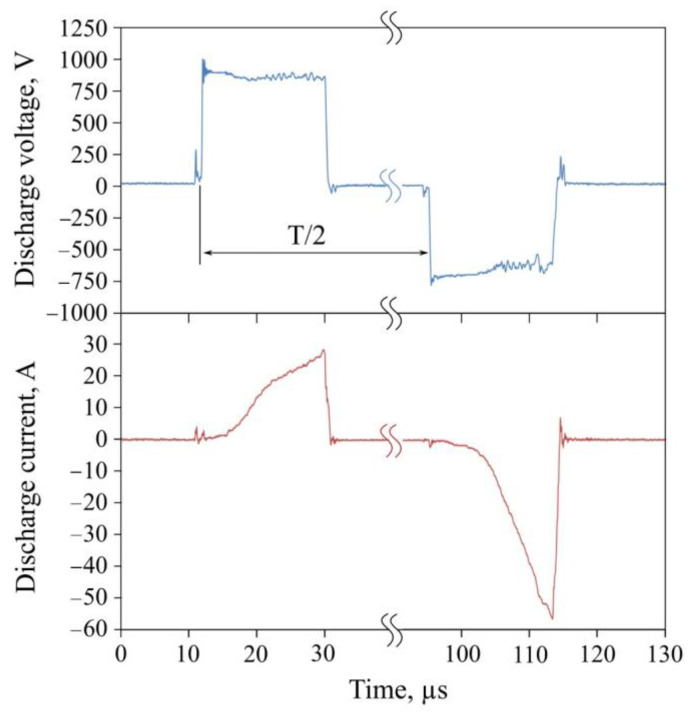
Oscilloscope patterns of the current and voltage waveforms of magnetrons with Ti and Al cathodes in the sputtering of the TiAlN coating.

**Figure 3 materials-16-01348-f003:**
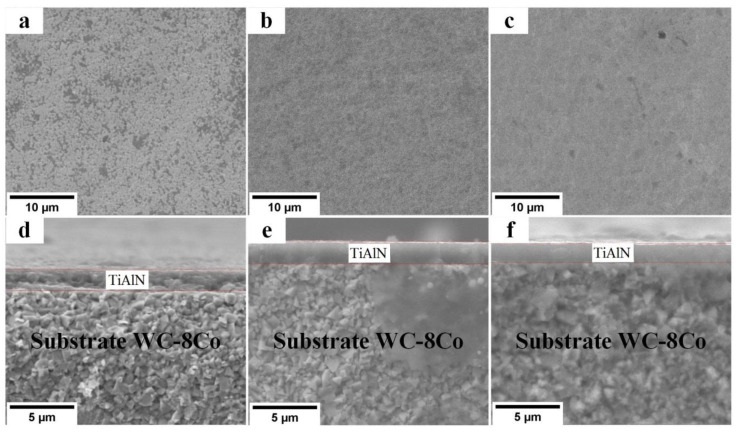
SEM images of the TiAlN coating surfaces and cross-sections deposited at the floating potential (**a**,**d**) and bias voltages of (**b**,**e**) −50 V and (**c**,**f**) −100 V. Al magnetron power: 30 W.

**Figure 4 materials-16-01348-f004:**
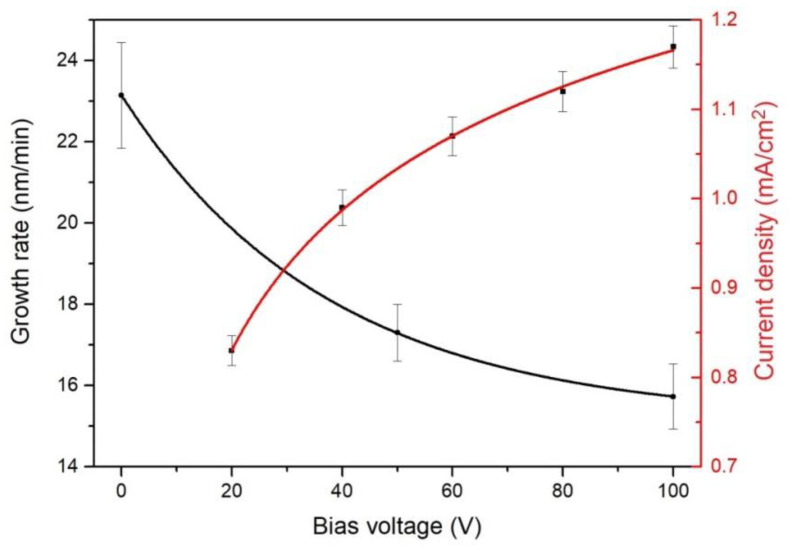
Dependences of deposition rate and substrate current density on the bias voltage.

**Figure 5 materials-16-01348-f005:**
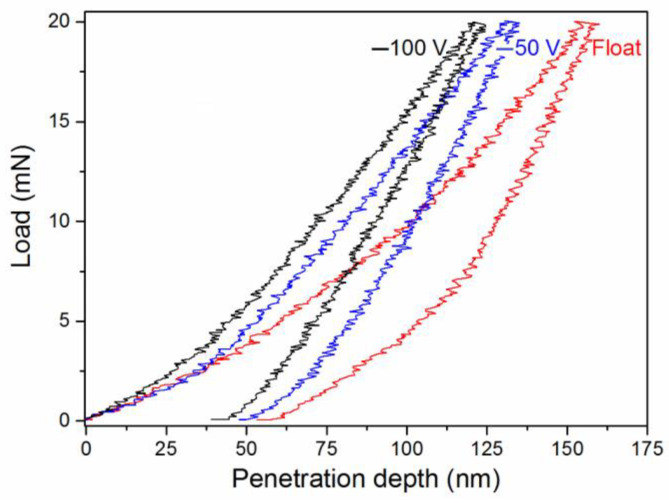
Loading–unloading curves of nanoindentation for the TiAlN coating deposited by the Al magnetron at 30 W of power and different substrate bias voltages.

**Figure 6 materials-16-01348-f006:**
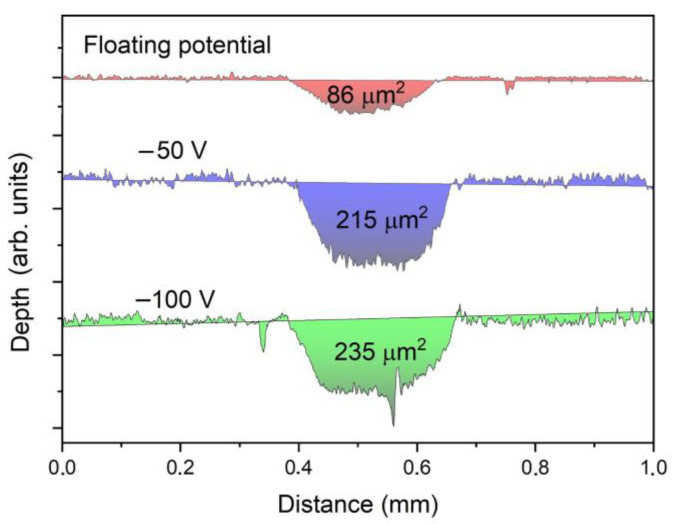
Wear track profiles of the TiAlN coatings deposited by the Al magnetron at 30 W of power and different substrate bias voltages.

**Figure 7 materials-16-01348-f007:**
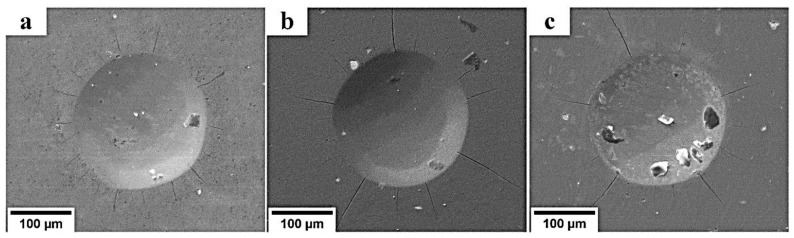
SEM images of indentation craters formed after the adhesion testing of the TiAlN-coated substrate surface: (**a**) at the floating potential and substrate bias voltages of (**b**) −50 V and (**c**) −100 V.

**Figure 8 materials-16-01348-f008:**
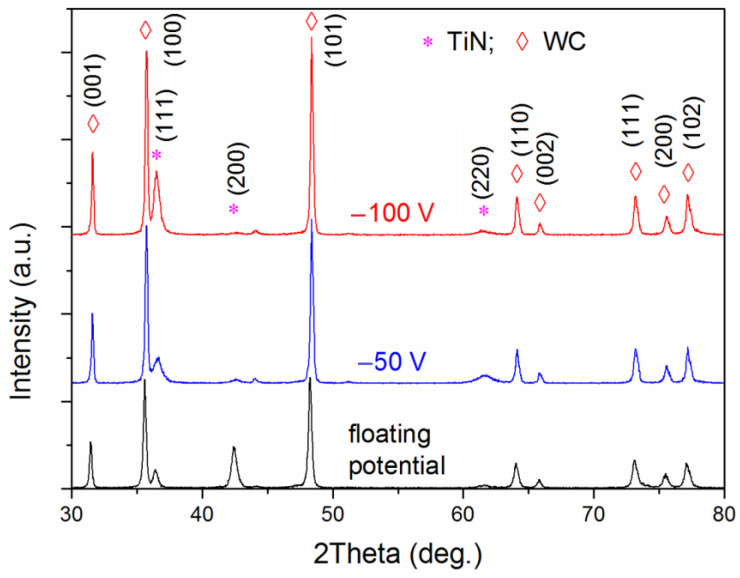
XRD patterns of the TiAlN coatings deposited by the Al magnetron at 30 W of power and different substrate bias voltages.

**Figure 9 materials-16-01348-f009:**
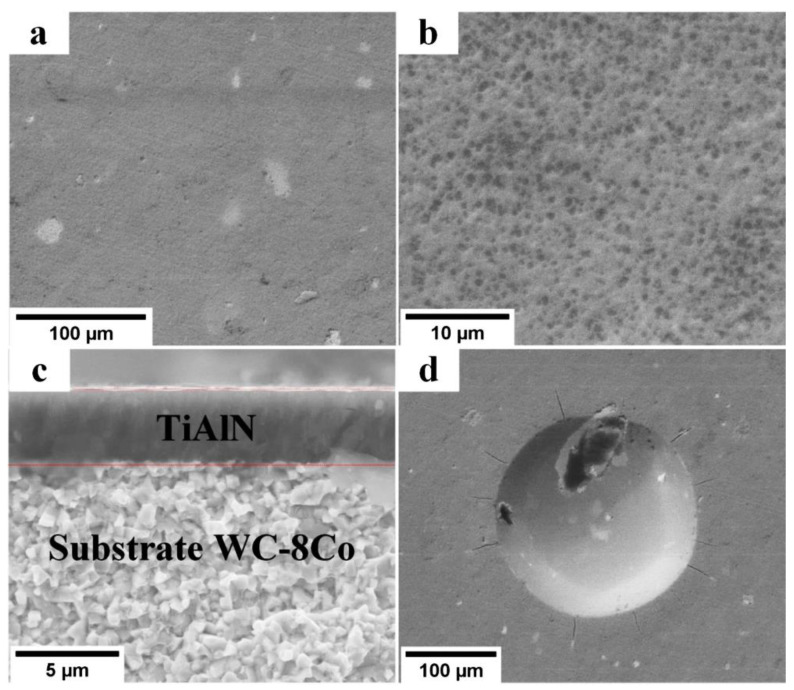
SEM images of the dual-HiPIMS-deposited Ti_0.61_Al_0.39_N coating: (**a**,**b**) surface, (**c**) cross-section, and (**d**) indentation crater formed after adhesion testing.

**Figure 10 materials-16-01348-f010:**
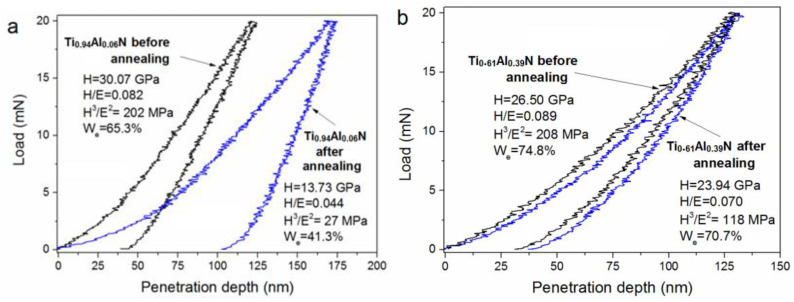
Loading–unloading curves obtained after the nanoindentation of the Ti_0.94_Al_0.06_N (**a**) and Ti_0.61_Al_0.39_N (**b**) coatings before and after annealing at 700 °C.

**Figure 11 materials-16-01348-f011:**
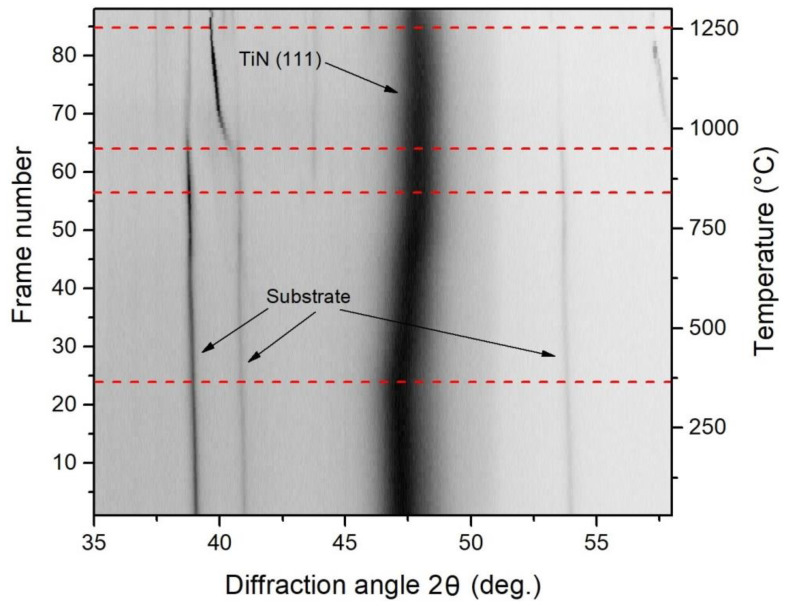
SXRD of the phase composition of the Ti_0.61_Al_0.39_N coating during heating to 1300 °C in air. λ = 0.172 nm.

**Figure 12 materials-16-01348-f012:**
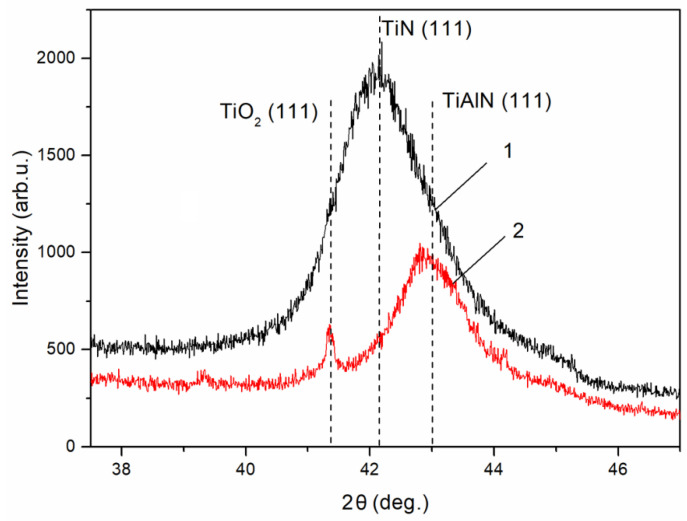
SXRD patterns of the Ti_0.61_Al_0.39_N coating obtained before (1) and after (2) heating to 1300 °C.

**Table 1 materials-16-01348-t001:** Elementary composition of TiAlN coatings deposited with an Al magnetron at 30 W of power and different bias voltages.

Substrate Biasing	Ti, at.%	Al, at.%	W, at.%	Co, at.%	N, at.%	(Ti+Al)/N	Chemical Composition
Floating potential	53.4	3.9	4.6	1.9	36.2	1.58	Ti_0.93_Al_0.07_N
−50 V	50.4	4.5	4.0	1.5	39.6	1.38	Ti_0.92_Al_0.08_N
−100 V	48.9	3.0	4.7	1.8	41.7	1.24	Ti_0.94_Al_0.06_N

**Table 2 materials-16-01348-t002:** Mechanical and tribological properties of the TiAlN coatings deposited by the Al magnetron at 30 W of power.

Substrate Biasing	*H*, GPa	*E*, GPa	*H*/*E*	*H*^3^/*E*^2^, MPa	*a*, µm	*P*_*max*_, GPa	*k*, mm^3^·N^−1^·m^−1^
Floating potential	17.01 ± 1.7	297.1 ± 17.2	0.057	56	80	1.49	1.8∙10^−6^
−50 V	25.65 ± 2.16	333.4 ± 18.9	0.077	152	79	1.55	4.5∙10^−6^
−100 V	30.07 ± 2.83	366.7 ± 11.6	0.082	202	77	1.60	4.9∙10^−6^

**Table 3 materials-16-01348-t003:** Elemental composition of the TiAlN coating deposited by the Al magnetron at 1 kW of power.

Ti, at.%	Al, at.%	W, at.%	Co, at.%	N, at.%	(Ti+Al)/N	Chemical Composition
24.6	15.6	11.0	3.9	45.1	0.89	Ti_0.61_Al_0.39_N

**Table 4 materials-16-01348-t004:** Mechanical and tribological properties of the Ti_0.61_Al_0.39_N coating deposited by the Al magnetron at 1 kW of power.

Substrate Biasing	*H*, GPa	*E*, GPa	*H*/*E*	*H*^3^/*E*^2^, MPa	*a*, µm	*P*_*max*_, GPa	*k*, mm^3^·N^−1^·m^−1^
−100 V	26.5 ± 1.11	298.9 ± 6.8	0.089	208	80	1.49	5.2·10^−6^

## Data Availability

Data sharing not applicable.

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
