# Peer review of "Properties of TiAlN Coatings Obtained by Dual-HiPIMS with Short Pulses"

_materials, 2023, doi:10.3390/ma16041348_

Round 1
Reviewer 1 Report
Page 1, Line 34: "Nevertheless, it is difficult to completely avoid the droplet fraction." The authors should claim on disadvantage of the droplets on the Arc PVS coating surfaces.
Page 2, Line 80-81: "...that has a posi-80 tive effect on the coating properties." The authors should clarify the word coating properties in the text.
Page 3, Line 100: The word vacuumized should be replaced with "evacuated".
Figure 1 can be improved by changing colors.
Page 4, Line 154-155:" ...5N normal load applied to 154 the Al2O3 ball of diameter 6 mm used as a counter body, and 5 mm wear scar."
The reason for selecting such a low load for harder coatings should be explained in the text. Moreover, selection of Al2O3 ball as a counter surface should be clarified in the text. Furthermore, one load is not enough to characterize wear resistance of the super hard coatings. Therefore, the 10 N and 15 N loads should be added. Additionally, total sliding distance was not given in the text. This important test parameter shoudl be given in the text. The Hertzian contact pressures of each load should be calculated and given in the paper.
Page 5, Line 159-160: The authors declared that they calculated the cross-sectional area of the wear scar by using Origin 9. This claim is needed to the clarify, on the other hand, there is no any evidence or illustration of wear scars in the paper.
It is hard to obtain coating thickness in Fig. 3. To improve paper quality the authors must extracted a sample and polished to evaluate cross-section. After that SEM/EDX analysis should be performed and cross-sections mesurements should be shown clearly for the readers on the SEM images.
Page 9, Line 256: "The lattice parameter a of the TiN coating deposited at the substrate floating..."
The authors did not mentioned how did they evaluate the lattice parameter.
Page 9, Line 265: "...which when subjected to high-temperature oxidation,..."
The word "when" should be removed from the statement.
The authors gave the elemental compositions of the coatings in Table 3, however, they did not discuss stoichiometry of the coatings.
The authors decleared that they measured the wear volume by profilometer and cross-sectional area. However, they did not show any wear scar or wear scar depth measurement results. They should give images from the wear depth evaluations and show the wear depth differences with wear depth profiles.
Archard's wear rate formula should be given in the text for wear rate presentations.
The novel and basic findings of the works should be imphesized in the conclusion section clearly.
Author Response
Page 1, Line 34: "Nevertheless, it is difficult to completely avoid the droplet fraction." The authors should claim on disadvantage of the droplets on the Arc PVS coating surfaces.
Response: The following text and new references 9,10 have been added to the revised paper: “A major shortcoming of CAE is the droplet fraction, which has a negative effect on the coating topography, the corrosion and oxidation stability (droplets act as diffusion paths), and the tribological properties [4, 7–9]. Besides, droplets could contribute to coating degradation by providing nucleation sites for shear cracks and by the release of abrasive fragments into the sliding contact [10].”
Page 2, Line 80-81: "...that has a positive effect on the coating properties." The authors should clarify the word coating properties in the text.
Response: Phrase “Secondly, a closed-field configuration of the dual system allows increasing the current density on the substrate, that has a positive effect on the coating properties.” has been changed on “Secondly, a closed-field configuration of the dual system allows increasing the current density on the substrate [25], that has a positive effect on the coating properties (adhesion, density and so on) [26, 27].”. New references 25–27 have been added.
Page 3, Line 100: The word vacuumized should be replaced with "evacuated".
Response: The world has been replaced.
Figure 1 can be improved by changing colors.
Response: The Figure 1 has been improved.
Page 4, Line 154-155:" ...5N normal load applied to the Al2O3 ball of diameter 6 mm used as a counter body, and 5 mm wear scar."
The reason for selecting such a low load for harder coatings should be explained in the text. Moreover, selection of Al2O3 ball as a counter surface should be clarified in the text. Furthermore, one load is not enough to characterize wear resistance of the super hard coatings. Therefore, the 10 N and 15 N loads should be added. Additionally, total sliding distance was not given in the text. This important test parameter should be given in the text. The Hertzian contact pressures of each load should be calculated and given in the paper.
Response: According to the literature (Table 1), the range of loads used in tribological testing of nitride coatings is 0.5–10 N. The choice of the middle value (5 H) was due to the desire to demonstrate the friction process and to eliminate the influence of adhesion on the obtained result. At higher loads, the cutting process will dominate and the adhesion between the coating and the substrate will affect the test result. A load of 5 N has also been used in [Ref. 6–9] (Table 1).
Various materials such as Al2O3, Si3N4, SiC, 100Cr6 steel are used as counterbodies (Table 1). When steel balls are used, the wear rate of the counterbody is usually significantly higher than that of the coating. This is due to the low hardness of the steel counterbody. To adequately estimate the wear rate of TiN and TiAlN coatings, we used an Al2O3 counterbody. It has high hardness and used by other researchers [Ref. 1,7–10] (Table 1).
Table 1.
â„– |
Reference |
Ball |
Load, N |
1 |
Huang, X.; Etsion, I.; Shao, T. Effects of elastic modulus mismatch between coating and substrate on the friction and wear properties of TiN and TiAlN coating systems. Wear 2015, 338–339, 54–61. doi:10.1016/j.wear.2015.05.016 |
sapphire |
0.5 |
2 |
Fan, Q.X.; Zhang, J.J.; Wu, Z.H.; Liu, Y.M.; Zhang, T.; Yan, B.; Wang, T.G. Influence of Al Content on the Microstructure and Properties of the CrAlN Coatings Deposited by Arc Ion Plating. Acta Metall. Sin. (Engl. Lett.) 2017, 30, 1221–1230. |
100Cr6 steel |
2 |
3 |
Kang, M.C.; Je, S.K.; Kim, K.H.; Shin, B.S.; Kwon, D.H.; Kim, J.S. Cutting performance of CrN-based coatings tool deposited by hybrid coating method for micro drilling applications. Surf. Coat. Technol. 2008, 202, 5629–5632. |
steel |
1 |
4 |
M-R. Alhafian, J-B. Chemin, Y. Fleming, L. Bourgeois, M. Penoy, R. Useldinger, F. Soldera, F. Mücklich, P. Choquet, Comparison on the structural, mechanical and tribological properties of TiAlN coatings deposited by HiPIMS and Cathodic Arc Evaporation, Surface and Coatings Technology, Volume 423, 2021, 127529. |
steel |
2 |
5 |
Xingrun, R.; Zhu, H.; Meixia, L.; Jiangao, Y.; Hao, C. Comparison of Microstructure and Tribological Behaviors of CrAlN and CrN Film Deposited by DC Magnetron Sputtering. Rare Metal Mat. Eng. 2018, 47, 1100–1106. |
Si3N4 |
3 |
6 |
Hsiao, Y.C.; Lee, J.W.; Yang, Y.C.; Lou, B.S. Effects of duty cycle and pulse frequency on the fabrication of AlCrN thin films deposited by high power impulse magnetron sputtering. Thin Solid Film. 2013, 549, 281–291. |
WC–6%Co |
5 |
7 |
Drnovšek, A.; Rebelo de Figueiredo, M.; Vo, H.; Xia, A.; Vachhani, S.J.; Kolozsvári, S.; Hosemann, P.; Franz, R. Correlating high temperature mechanical and tribological properties of CrAlN and CrAlSiN hard coatings. Surf. Coat. Technol. 2019, 372, 361–368. |
Al2O3 |
5 |
8 |
Ramadoss, R.; Kumar, N.; Pandian, R.; Dash, S.; Ravindran, T.R.; Arivuoli, D.; Tyagi, A.K. Tribological properties and deformation mechanism of TiAlN coating sliding with various counterbodies. Tribol. Int. 2013, 66, 143–149. |
100Cr6 steel, SiC, Al2O3 |
5 |
9 |
Sidelev, D.V.; Voronina, E.D.; Kashkarov, E.B. Duplex treatment of AISI 420 steel by RF-ICP nitriding and CrAlN coating deposition: The role of nitriding duration. Coatings 2022, 12, 1709. |
Al2O3 |
5 |
10 |
G.S Kim, B.S. Kim, S.Y. Lee, J.H. Hahn. Effect of Si content on the properties of TiAl–Si–N films deposited by closed field unbalanced magnetron sputtering with vertical magnetron sources, Thin Solid Films 506– 507 (2006) 128 – 132. |
Al2O3 |
10 |
The sliding distance during the tribological tests was 150 meters.
A maximum initial Hertzian contact pressure has been calculated for obtained TiAlN coatings. On the assumption that the Poisson's ratio for TiAlN coatings is 0.177 [De Hosson, J.T.M., Carvalho, N.J.M., Pei, Y., Galvan, D. (2006). Electron Microscopy Characterization of Nanostructured Coatings. In: Cavaleiro, A., De Hosson, J.T.M. (eds) Nanostructured Coatings. Nanostructure Science and Technology. Springer, New York, NY. https://doi.org/10.1007/978-0-387-48756-4_5].
To calculate the Hertzian contact pressure we used a modulus of elasticity of counterbody of 382 GPa and a Poisson's ratio of 0.24 [Ramadoss, R.; Kumar, N.; Pandian, R.; Dash, S.; Ravindran, T.R.; Arivuoli, D.; Tyagi, A.K. Tribological properties and deformation mechanism of TiAlN coating sliding with various counterbodies. Tribol. Int. 2013, 66, 143–149.].
The calculated values of Hertzian contact pressure have been added to Tables 2 and 4 of the revised paper.
Page 5, Line 159-160: The authors declared that they calculated the cross-sectional area of the wear scar by using Origin 9. This claim is needed to the clarify, on the other hand, there is no any evidence or illustration of wear scars in the paper.
Response: We added wear tracks profiles of coatings in the revised paper (Fig. 6).
It is hard to obtain coating thickness in Fig. 3. To improve paper quality the authors must extracted a sample and polished to evaluate cross-section. After that SEM/EDX analysis should be performed and cross-sections mesurements should be shown clearly for the readers on the SEM images.
Response: The coating thickness was determined by Calotest technique using a CAT-S-0000 Calotest machine (CSEM, Switzerland). Therefore, SEM images were not used to determine the thickness of the coating. We agree that the SEM images are not of very good quality, but even at this quality they allow one to see the difference in the density of the coating structure. Unfortunately, it is not possible to get better images in a short period of time. SEM/EDX analysis of the coatings was made and given in the article.
Page 9, Line 256: "The lattice parameter a of the TiN coating deposited at the substrate floating..."
The authors did not mentioned how did they evaluate the lattice parameter.
Response: The lattice parameter a was determined using the XRD method from d-spacing, and the structural lattice parameters were refined using the Rietveld method.
Page 9, Line 265: "...which when subjected to high-temperature oxidation,..."
The word "when" should be removed from the statement.
Response: Corrected
The authors gave the elemental compositions of the coatings in Table 3, however, they did not discuss stoichiometry of the coatings.
Response: We revised the description of Table 3. For example, the sentence has been added: “The increase in nitrogen content in the coating is due to the more efficient capture of nitrogen atoms in the near-surface regions of the coating and their more intense excitation and ionisation, stimulating the chemical reaction of nitride formation.”
The authors decleared that they measured the wear volume by profilometer and cross-sectional area. However, they did not show any wear scar or wear scar depth measurement results. They should give images from the wear depth evaluations and show the wear depth differences with wear depth profiles.
Response: We added wear tracks profiles of coatings in the revised paper (Fig. 6).
Archard's wear rate formula should be given in the text for wear rate presentations.
Response: We describe the method for determining the wear rate in section 2.2: “The wear rate resulted from the ratio between the wear volume and the product of normal force and total sliding distance, viz. mm3·N–1·m–1.”
The novel and basic findings of the works should be emphasized in the conclusion section clearly.
Response: The novelty of work was emphasized in the conclusion.
Reviewer 2 Report
My comments are technical.
- a more detailed description of table 2 is required,
- Figures 5 and 9 should be changed, Indentation curves are usually presented as lines, not as outlined areas.
Author Response
My comments are technical.
- a more detailed description of table 2 is required,
Response: Some comments added in description of table 2.
- Figures 5 and 9 should be changed, Indentation curves are usually presented as lines, not as outlined areas.
Response: Figures 5 and 9 have been changed according Reviewer suggestion.
Reviewer 3 Report
This manuscript deals with the growth and mechanical properties of TiN and TiAlN films grown by using dual-HiPIMS configuration. The investigation is generally of great interest to the hard coating community. It is recommended for publication after the following questions are addressed.
(1) Section 2.1 Preparation of TiAlN coatings: The detailed deposition parameters for the TiN series and TiAlN series are not addressed, for example, the bias voltage for the TiN series was altered. In addition, I suggest changing the title of section 2.1 as both TiN and TiAlN are deposited.
(2) Section 2.1, paragraph 2: “The TiN coating was deposited using the dual magnetron. Al and Ti cathodes, both 98 of 99.95% purity and a diameter 76 mm were used for sputtering”. I am not sure if the discharge of Al would generate the incorporation of Al in the TiN film.
(3) Section 2.1, paragraph 2: “Prior to the TiN coating deposition, the bipolar voltage with negative pulses of 900 V, 100 kHz, and 70% duty cycle, was supplied to the substrate to improve the adhesive strength and provide a 5-min surface activation. Afterwards, the negative pulse amplitude reduced to 100 V”. If bipolar substrate voltage was used for substrate etching and deposition, the positive pulse amplitude should be addressed.
(4) I am wondering if the cooling system could stand such a high power (1 kW) for the 3-inch target. Maybe it is not a problem as the peak current is not extremely high.
(5) Section 3.1: “One can see that the thickness grows with increasing bias 176 voltage and, consequently, the energy of bombarding ions.” Please could you explain why the film deposition rate increases with the increasing ion-bombarding energy? This phenomenon is not in line with the other research as the higher bombarding energy generates higher resputtering. And this part is conflict with the data shown in Fig. 4.
(6) It is difficult to follow the discussion on the change of nitrogen elemental composition. Why does the nitrogen ionization probability change with the varied substrate bias voltage at the same target discharge condition? The same question appears in the “substrate current density” (is it ion current density or the substrate current density which includes the electron current density? Please clarify) in Fig. 4, why the substrate bias voltage increases the “substrate current density”?
(7) Fig. 7: To the best of my knowledge, the (100) plane for TiN possesses a lower surface energy compared to that of the (111) plane, which should be compared with the lower surface energy of the (111) plane for fcc-structured metallic films. Please double-check the results and discussion of this part.
Author Response
Reviewer 3.
This manuscript deals with the growth and mechanical properties of TiN and TiAlN films grown by using dual-HiPIMS configuration. The investigation is generally of great interest to the hard coating community. It is recommended for publication after the following questions are addressed.
(1) Section 2.1 Preparation of TiAlN coatings: The detailed deposition parameters for the TiN series and TiAlN series are not addressed, for example, the bias voltage for the TiN series was altered. In addition, I suggest changing the title of section 2.1 as both TiN and TiAlN are deposited.
Response: The title of section 2.1 has not been changed according Reviewer suggestion because we refined the chemical composition of TiN coatings and they become TiAlN coatings with small Al content. Some details of coating deposition were added.
(2) Section 2.1, paragraph 2: “The TiN coating was deposited using the dual magnetron. Al and Ti cathodes, both 98 of 99.95% purity and a diameter 76 mm were used for sputtering”. I am not sure if the discharge of Al would generate the incorporation of Al in the TiN film.
Response: Yes, you are right. The composition of the films has been updated and new data has been added to the revised paper. TiN coating contains a meaningfull amount of Al (Table 1 of revised paper). Therefore we have replaced the designation TiN with TiAlN throughout the article.
(3) Section 2.1, paragraph 2: “Prior to the TiN coating deposition, the bipolar voltage with negative pulses of 900 V, 100 kHz, and 70% duty cycle, was supplied to the substrate to improve the adhesive strength and provide a 5-min surface activation. Afterwards, the negative pulse amplitude reduced to 100 V”. If bipolar substrate voltage was used for substrate etching and deposition, the positive pulse amplitude should be addressed.
Response: Positive pulse has duration of 4 μs and peak amplitude, which was 20% of the negative pulse.
(4) I am wondering if the cooling system could stand such a high power (1 kW) for the 3-inch target. Maybe it is not a problem as the peak current is not extremely high.
Response: 1 kW is not extremely high power and does not require special design of the cooling system.
(5) Section 3.1: “One can see that the thickness grows with increasing bias voltage and, consequently, the energy of bombarding ions.” Please could you explain why the film deposition rate increases with the increasing ion-bombarding energy? This phenomenon is not in line with the other research as the higher bombarding energy generates higher resputtering. And this part is conflict with the data shown in Fig. 4.
Response: Thank you. We missed this error during the translation of the paper. The phrase should read as follows: “One can see that the coating density grows with increasing bias voltage and, consequently, energy of bombarding ions.” Of course, the film deposition rate decreases with the increasing ion-bombarding energy, as shown on Fig. 4.
(6) It is difficult to follow the discussion on the change of nitrogen elemental composition. Why does the nitrogen ionization probability change with the varied substrate bias voltage at the same target discharge condition? The same question appears in the “substrate current density” (is it ion current density or the substrate current density which includes the electron current density? Please clarify) in Fig. 4, why the substrate bias voltage increases the “substrate current density”?
Response: We deleted the questionable phrase: "Kong et al. [28] report on the increase in the N concentration from 50 to 56 at.% in depositing CrN films by medium frequency magnetron sputtering with increasing bias voltage from −100 to −300 V, which is attributed to the increase in ionized N atoms.”
Yes, at small bias voltages the current to the substrate includes electron and ion current. When the bias voltage is smoothly increased, the electron current to the substrate decreases and the ion current increases. At a voltage of about -100 V, the current to the substrate goes into saturation, as all the ions from the nearby plasma region are collected by the substrate.
(7) Fig. 7: To the best of my knowledge, the (100) plane for TiN possesses a lower surface energy compared to that of the (111) plane, which should be compared with the lower surface energy of the (111) plane for fcc-structured metallic films. Please double-check the results and discussion of this part.
Response: Yes, you are right. M. Lattemann [Fully dense, non-faceted 111-textured high power impulse magnetron sputtering TiN films grown in the absence of substrate heating and bias. Thin Solid Films 518 (2010) 5978–5980] writes: “Texture evolves toward 111, even though the 001 surface energy is lower [12], due to kinetic limitations associated with the higher cation surface
diffusivity on TiN(001) than TiN(111)”. The text of the paper has been amended.
Round 2
Reviewer 1 Report
I have check the review responds and decided that the authors had shown great effort to improve the manuscript. Congrulations!